# Neural manifold under plasticity in a goal driven learning behaviour

**Barbara Feulner**, **Claudia Clopath** *

Department of Bioengineering, Imperial College London, London, United Kingdom

* c.clopath@imperial.ac.uk

## Abstract

Neural activity is often low dimensional and dominated by only a few prominent neural covariation patterns. It has been hypothesised that these covariation patterns could form the building blocks used for fast and flexible motor control. Supporting this idea, recent experiments have shown that monkeys can learn to adapt their neural activity in motor cortex on a timescale of minutes, given that the change lies within the original low-dimensional subspace, also called neural manifold. However, the neural mechanism underlying this within-manifold adaptation remains unknown. Here, we show in a computational model that modification of recurrent weights, driven by a learned feedback signal, can account for the observed behavioural difference between within- and outside-manifold learning. Our findings give a new perspective, showing that recurrent weight changes do not necessarily lead to change in the neural manifold. On the contrary, successful learning is naturally constrained to a common subspace.

## Author summary

It has been suggested that the coordinated activation of neurons might play an important role for movement execution. Whether such activity patterns are fixed or flexibly relearned remains matter of debate. It has been shown that monkeys can learn within minutes to adjust their neural activity, as long as they use the initial set of activity patterns. In contrast, monkeys needed several days and a sequential training procedure to learn completely new patterns. Here, we developed a computational model to investigate which biological features might lead to these experimental observations. Learning in our model is implemented through weight changes between neurons in a recurrently connected network. In order for these weight changes to improve the produced behaviour, an error signal is required which tells each neuron whether it should increase or decrease its activity in order to produce a movement closer to the target movement. We found that learning such an error signal is possible only in the first experimental condition, where monkeys needed to adapt their neural activity using already existing activity patterns. The learning of this error signal therefore poses a major constraint on what type of changes in neural activity can and can not be learned.

**Data Availability Statement:** All code is available on GitHub (https://github.com/babaf/neural-manifold-and-plasticity.git).

**Funding:** This work has been funded by BBSRC (BB/N013956/1 and BB/N019008/1), Wellcome Trust (200790/Z/16/Z), the Simons Foundation

(564408), and the EPSRC (EP/R035806/1) (all to CC). The funders had no role in study design, data collection and analysis, decision to publish, or preparation of the manuscript.

**Competing interests:** The authors have declared that no competing interests exist.

## Introduction

The dynamics of single neurons within a given brain circuit are not independent, but highly correlated. Moreover, neural activity is often dominated by only a very low number of distinct correlation patterns [1–15]. This implies that there exists a low-dimensional manifold in the high-dimensional population activity space, to which most of the variance of the neural activity is confined. Why such low-dimensional dynamics is observed in the brain remains unclear. One potential reason is that most experimental designs are inherently low-dimensional, and therefore bias the observed neural activity [16]. Alternatively, it has been shown that low-dimensional dynamics can arise from structured connectivity within the network [17–25].

The functional implications of low-dimensional neural dynamics are also unclear [26, 27]. For example, preparatory and movement activity lie in orthogonal subspaces, which can serve as a gating mechanism for movement initiation [3, 28]. Furthermore, the same manifold might underlie multiple skilled wrist and reach-to-grasp movements [29], as well as the same task over a long timescale [30]. These results suggest that there is one stable manifold for motor control, and that learning and performing different, but similar, movements are happening within this scaffold [11].

Using a brain-computer interface (BCI) [31] in monkeys, Sadtler et al. showed that it is possible to adapt neural activity in motor cortex, if the new activity pattern lies within the original manifold, but not if it lies outside of it [32]. They could test this by introducing two kinds of perturbations to a BCI, which was previously trained to decode two-dimensional cursor dynamics from neural activity. The monkeys had to adapt their neural activity in motor cortex to produce the correct cursor dynamics, given the perturbed BCI mapping. Interestingly, the timescale of within-manifold learning is in the order of minutes to hours [32]. In contrast, outside-manifold learning is not possible within a single session, but instead requires progressive training over multiple days [33]. Due to the timescale difference, it has been hypothesized that within-manifold learning is related to fast motor adaption and outside-manifold learning is related to skill learning. Yet, the underlying neural mechanisms for both types of learning remain unknown.

Previously, two computational studies aimed at uncovering the mechanisms underlying the constraints on learning in the described experiment. Waernberg and Kumar used a simplified implementation of the BCI task to compare the amount of weight change which is required for within- and outside-manifold learning. They found that within-manifold learning requires less weight change in their simulations and thereby concluded that within-manifold learning is faster because of smaller underlying synaptic changes [34]. In contrast, Menendez and Latham investigated the possibility that learning results from a change in input to motor cortex [35]. Although there have been these two attempts to explain the experimental observations in Sadtler et al., the search for underlying neural mechanisms remains inconclusive.

We used computational modelling to study the relationship between neural manifold and learning. As motor learning can drive network rewiring in motor cortex on a short timescale [36–39], we wanted to test whether local network rewiring can account for within-manifold, as well as outside-manifold learning. To that end, we implemented an in-silico version of the BCI experiment in Sadtler et al., where motor cortex activity is simulated by a recurrent neural network (RNN) [17, 18, 24]. We showed that an ideal observer training algorithm is able to learn within- and outside-manifold perturbations equally well. One component of the algorithm is that errors in the produced cursor dynamics are translated to errors in single neuron activity. This form of feedback signal is used to adapt the local connections within the network in order to minimize the observed error. As the correct feedback signal is not a priori given in the biological system, we subsequently tested more realistic feedback scenarios. By disrupting the

correct feedback signal, we found that within-manifold learning is more robust to erroneous or sparse feedback signals. Furthermore, learning the feedback signal from scratch was only possible for within-manifold perturbations. To enable feedback learning for outside-manifold perturbations we had to introduce an incremental strategy, which resembles the progressive training used in experiments [33].

# Results

## Recurrent neural network (RNN) performing a center-out-reach task

To investigate the potential difference between within- and outside-manifold learning, we implemented an in-silico version of a brain-computer interface (BCI) experiment previously done with monkeys [32]. Instead of measuring neural activity in monkey motor cortex, we implemented a recurrent neural network (RNN) which was trained to do a similar center-out-reach task as the monkey had to perform (Fig 1A). After the initial training phase, we perturbed the BCI mapping to either trigger a within- or outside-manifold change (Fig 1C and 1D). We then retrained the recurrent network. This pipeline allowed us to test to what extent reorganization in the local network can explain the behaviourally observed differences in within- and outside-manifold learning.

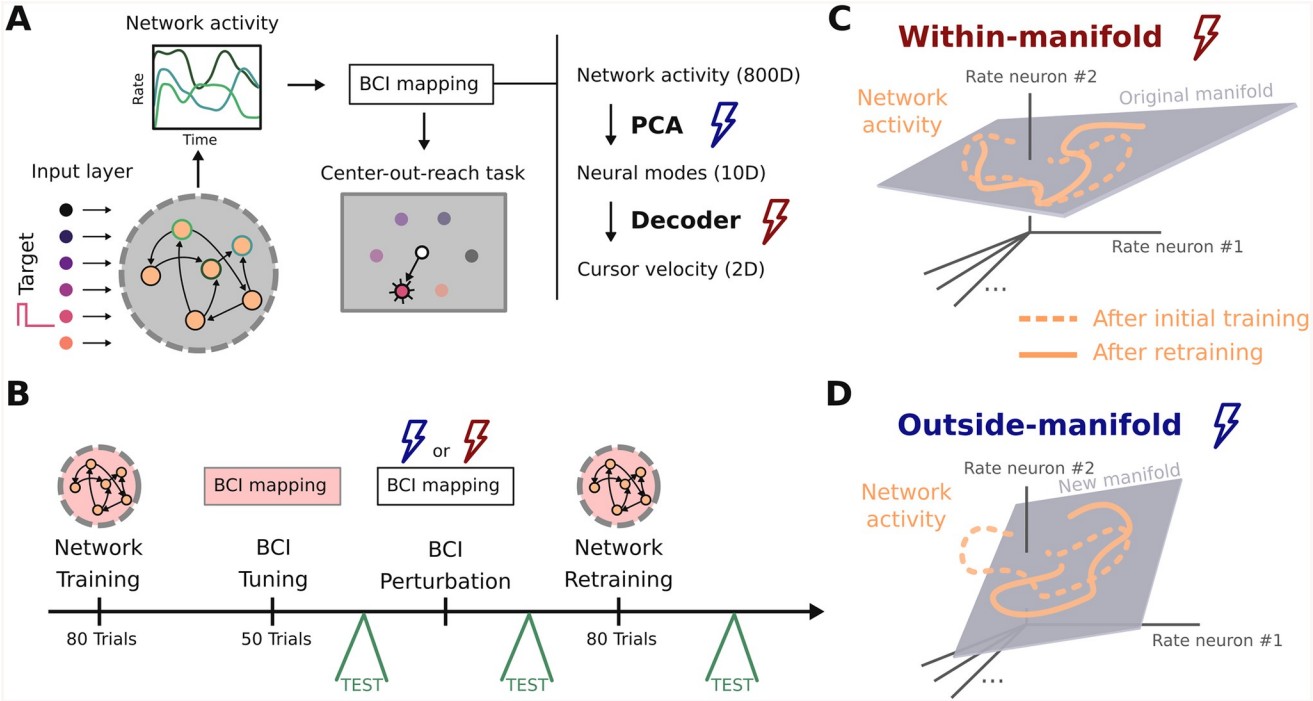

**Fig 1. Recurrent neural network (RNN) performing a center-out-reach task.** (A) In order to study learning within and outside of the neural manifold, we used a brain-computer interface (BCI) setup similar to [32]. A recurrent neural network serves as generator of network activity which is then used via a BCI setup to control a cursor in a two-dimensional center-out-reach task. The key step in our simulation is to perturb the initial BCI mapping and let the recurrent network learn to compensate for that. (B) The simulation design. (C and D) In order to compensate the BCI perturbation the network needs to produce new dynamics. For a within-manifold perturbation (C) the new target activity is constrained to the same manifold as the original activity. For an outside-manifold perturbation (D) this is not the case, but the network activity has to explore new dimensions in state space.

## RNN with correct feedback signal can learn within- and outside-manifold equally well

In order to understand constraints on within- and outside-manifold learning, we firstly simulated our network with an ideal-observer feedback signal. This means that the task error, which is defined as the difference between target and produced cursor velocity, is correctly assigned to errors on the activity of single neurons in the network. The correct credit assignment can be achieved by a matrix multiplication with the pseudo-inverse of the BCI readout. By doing this, we intentionally ignored the fact that a biological system would need to learn this credit assignment first, before it can be used for retraining. Rather, we first focused on the restructuring of the recurrent connections. With this training method, our network reached high performance in the center-out-reach task after the initial training phase (Fig 2A1). To probe within- versus outside-manifold learning, we then applied the corresponding perturbation to the BCI mapping, which led to impaired cursor trajectories in both cases (Fig 2A2). Interestingly, using the same training method as during the initial training phase, we could retrain the network to adapt to the changed BCI mapping for within-manifold perturbation, as well as for outside-manifold perturbation. The cursor trajectories after the initial training phase and after the retraining phase looked similar for both types of perturbations (Fig 2A3). We quantified the performance by measuring the mean squared error between target and produced cursor velocities (Fig 2B). The average retraining performance confirms that within- and outside-manifold perturbations can equally well be counteracted and that this result is not constrained to a specific network initialization (Fig 2B), task setup (S1, S5, S19 and S20 Figs), or learning algorithm (S3 and S6 Figs). Comparing learning timescales did not show any difference for within- versus outside-manifold learning (S16 Fig), yet a more biologically plausible learning algorithm showed slightly slower learning for outside-manifold perturbations (S6 Fig). Together, these results indicate that there is no fundamental difference in learning new dynamics either within the original manifold or outside of it, as long as we focus solely on the learning of the recurrent weights.

Given biological constraints on the amount of affordable synaptic restructuring, we wanted to test whether there is a difference in weight change needed to compensate either within- or outside-manifold perturbations. To investigate this, we started with comparing the network connectivity before and after the retraining phase (Fig 2C). We found that there is no difference in the amount of weight change produced by either within- or outside-manifold retraining. In fact, the weight change distribution for both types of retraining is similar to the distribution obtained by comparing the weights before and after the initial training phase. In addition to the amount of weight change, we also compared the effective rank and dimensionality of weight change between within- and outside-manifold relearning (S4 Fig). We found that weight changes happening during outside-manifold learning are slightly higher dimensional compared to weight changes during within-manifold learning. These results show that retraining within or outside of the original manifold does not necessarily lead to a different amount of weight change, yet weight changes related to within-manifold learning are slightly less dimensional.

In order to understand how our network achieves high performance after the adaptation phase, we analysed the underlying dynamics before and after adaptation. To quantify the dynamical changes, we recalculated the internal manifold after the retraining phase and compared it to the original manifold (Fig 2D). For within-manifold relearning, we found that the internal manifolds before and after relearning have high overlap as expected (Fig 2D red). Interestingly, learning is not equally distributed across all ten modes in the manifold, but concentrated on the most prominent ones (S7 Fig). For outside-manifold, we computed two

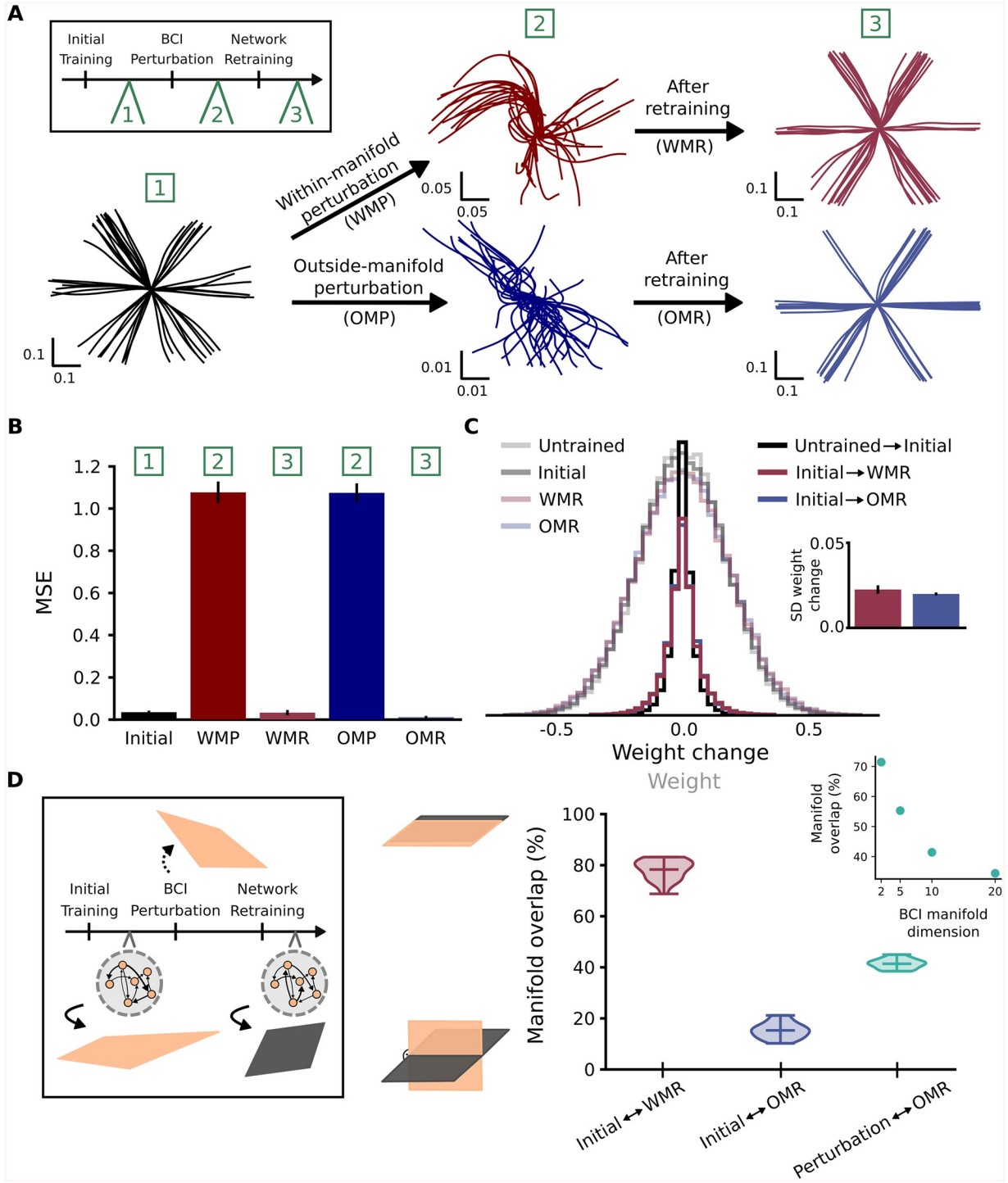

**Fig 2. RNN with correct feedback signal can learn within- and outside-manifold equally well.** (A) Cursor trajectories after initial training phase (on the left), after within- (WMP) or outside-manifold (OMP) perturbation (in the middle) and after within (WMR) or outside (OMR) retraining phase (on the right) for one example simulation. (B) Performance measured as mean squared error (MSE) between target cursor velocity and produced cursor velocity. Shown are the average simulation results for twenty randomly initialized networks, bars indicate the standard deviation across networks. (C) Weight distribution and weight change distribution during the experiment. The inlet shows the average standard deviation of weight change, bars indicate standard deviation across networks. (D) Manifold overlap between before and after retraining phase (red and blue), or between target manifold for outside-manifold perturbation and internal manifold after retraining phase (green). Inlet shows how the overlap between BCI target manifold and internal manifold depends on BCI manifold dimension.

measures: 1) how much the internal manifold after retraining still overlaps with the original one (Fig 2D blue), and 2) how the internal manifold after retraining has aligned to the one defined by the BCI perturbation (Fig 2D green). We found that although the task performance is high after retraining the network following an outside-manifold perturbation, the network dynamics do not completely align with the new readout defined by the altered BCI mapping. More detailed analysis showed that outside-manifold learning focuses on the few most prominent modes, which have initially the highest decoder values (S8 and S9 Figs). This demonstrates that it is not necessary that network dynamics completely realign with the altered BCI mapping to perform well in the task. Instead it is enough to have a sufficient amount of overlap, or match the most prominent modes. The amount of necessary overlap likely depends on a combination of task dimensionality [16] and BCI manifold dimensionality (Fig 2D inlet).

## RNN with corrupted feedback signal or constraint on weight change can learn better within- than outside-manifold

Our results so far do not show a difference between within- and outside-manifold learning, in constrast to experimental observations [32]. To further investigate where this behaviourally observed difference potentially comes from, we next dropped the assumption of an ideal-observer feedback signal, which is biologically highly implausible, and tested the effect of different corrupted feedback versions on recurrent retraining. As neural signals are noisy and long-range projections are sparse, we tested the effect of a noisy, as well as sparse feedback signal on relearning performance. We started by analysing the effect of a noisy feedback signal (Fig 3B). We simulated our experiment with different strengths of noise corruption in the feedback signal and compared the task performance after retraining. Interestingly, we found a difference between within- and outside-manifold learning, as within-manifold learning shows for a certain intermediate noise level better retraining performance than outside-manifold learning (Fig 3B top). This difference between within- and outside-manifold learning holds when we simulated the BCI experiment using a different learning rule to modify recurrent weights (S18 Fig). Next, we investigated the effect of a sparse feedback signal (Fig 3C). In this scenario, not all neurons in the recurrent network receive a feedback signal and therefore participate in the recurrent restructuring. We again found that within-manifold learning is achieving better performance with the same portion of feedback signals available (Fig 3C top). Finally, we tested the effect of sparse plastic weights within the recurrent network (Fig 3D). Also there we found that within-manifold learning is preferential. All together, these results show that within-manifold learning outperforms outside-manifold learning in the presence of either erroneous feedback signals, sparse feedback signals, or if there is a constrained number of plastic connections within the network.

Although the three different corrupted training scenarios show similar results with respect to retraining performance, the specific kind of disturbance could yet be highly different. Therefore, we analysed in more detail the network dynamics before and after retraining. We compared how well the internal manifold after retraining aligns with either the initial manifold in the case of within-learning, or with the dictated manifold coming from the BCI perturbation in the case of outside-learning. Interestingly, we found that, for noise in the feedback signal, the network dynamics are driven off the target manifold (Fig 3B center). This makes sense as the feedback signal is disturbed in random directions. During the retraining phase, the network then tries to align its internal manifold to these random directions, resulting in reduced overlap to the correct manifold dimensions. We found that task performance is high as long as manifold overlap is more than 40% (Fig 3B bottom), which is in line with what we found for outside manifold learning with the correct feedback signal (Fig 2D). In contrast, reducing the

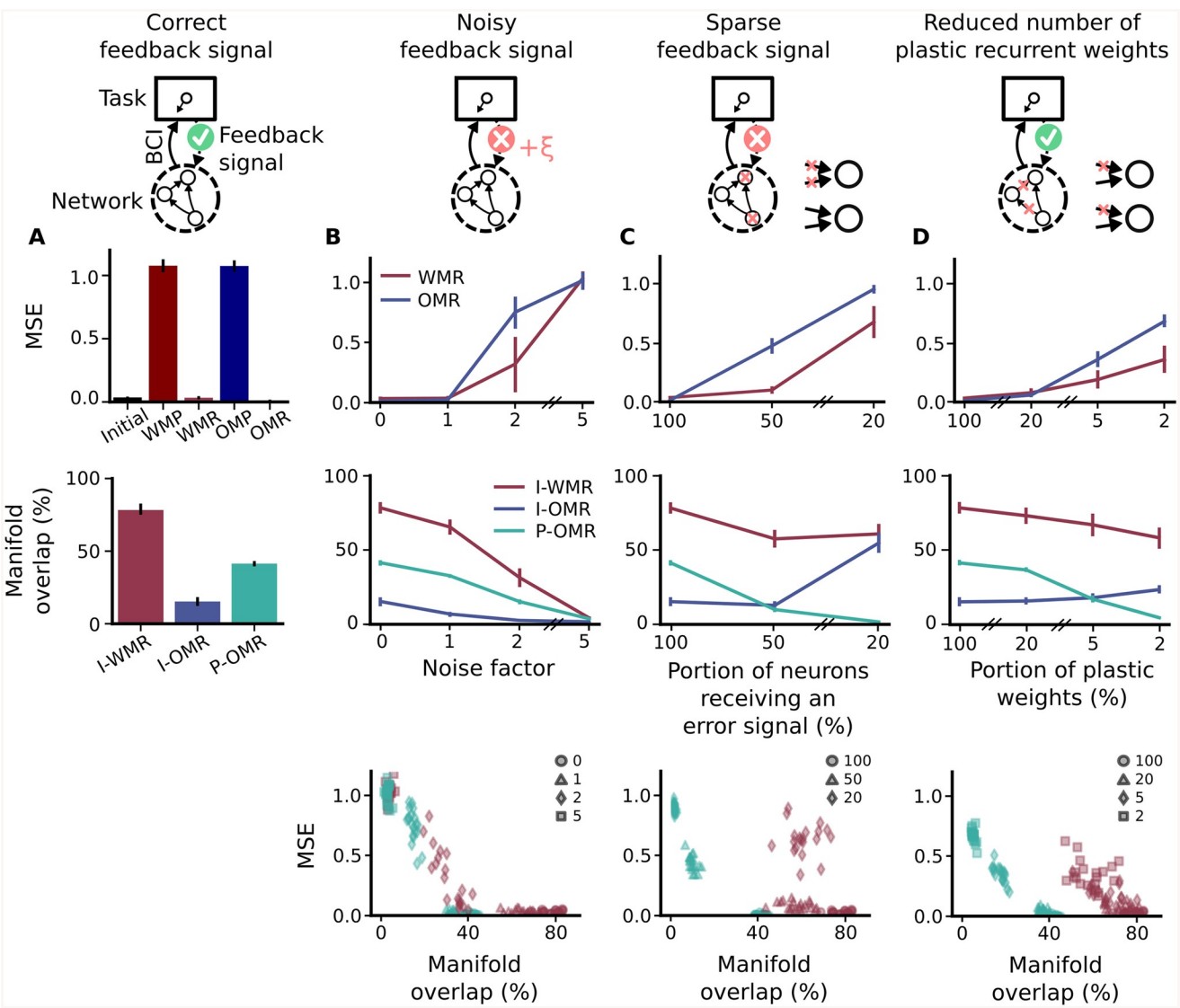

**Fig 3. RNN with corrupted feedback signal or constraint on weight change can learn better within- than outside-manifold.** (A) Relearning results for RNN with correct feedback signal (same as in Fig 2), bars indicate here and in the following standard deviation across networks. WMP and OMP show performance after within- or outside-manifold perturbations, whereas WMR and OMR show performance after retraining of the respective BCI perturbations (top). Manifold overlap is measured between initial manifold and manifold after within learning (I-WMR), initial manifold and manifold after outside learning (I-OMR) and manifold defined by BCI perturbation and manifold after outside learning (P-OMR) (center). (B) Relearning results for noisy feedback signal. The correct feedback transformation is distorted by adding independent noise to each entry of the matrix. The noise is drawn from a zero-mean Gaussian with standard deviation $\sigma = \alpha \cdot \sigma_{original}$, where $\alpha$ is the noise factor and $\sigma_{original}$ is the standard deviation of the correct feedback matrix. Top panel shows performance results, center panel shows manifold overlap and bottom panel shows the relation between both. (C) Relearning results for sparse feedback signal. In this scenario not all neurons in the recurrent network receive a feedback signal, which leads to a portion of recurrent weights remaining static during retraining. (D) Relearning results for sparse plastic connections. Similar to (C) a portion of recurrent weights remain static during retraining. But in contrast to (C), these static connections are not clustered to specific neurons. Instead, all neurons keep at least one plastic incoming connection.

number of plastic connections has a different effect on recurrent restructuring. It primarily disturbs the manifold alignment for outside-manifold perturbations (Fig 3C and 3D center). This is due to the fact that the static connections in the network constrain the dynamics to stay within the original manifold. Therefore, for within-manifold learning the manifold overlap

between before and after retraining is always relatively high (Fig 3C and 3D center). Nevertheless, the task performance is reduced in these cases as within-manifold perturbations can not correctly be counteracted (Fig 3C and 3D top). Furthermore, we found that not only the ratio of plastic connections affects retraining, but also the level of disturbance. As long as each neuron in the network receives a feedback signal, retraining performance is relatively high even if only 20% of the connections are plastic (Fig 3D top). In summary, although manifold overlap and task performance correlate in some regimes (e.g. Fig 3D bottom), we can find cases with high task performance and low manifold overlap (outside-manifold learning with correct feedback signal), and we can find cases with low task performance and high manifold overlap (within-manifold learning with sparse plastic connections), showing that one does not necessarily imply the other.

## Learning the feedback signal is possible for within- but not for outside-manifold perturbations

Although we have shown that within-manifold learning is more robust to corrupted feedback signals, the question remains of how the biological system could solve the error assignment and learn the feedback weights. A very naive hypothesis would be that it can infer the feedback weights by simply knowing its own internal dynamics and regressing them against the observed cursor dynamics (Fig 4A). We tested this hypothesis and surprisingly found that this simple strategy gives good estimates of the feedback weights in the case of a within-manifold perturbation (Fig 4B), but fails completely in the case of an outside-manifold perturbation (Fig 4C). To quantify the quality of feedback learning, we calculated the correlation coefficient between the learned and the correct feedback weights. We found that the feedback learning performance is worse compared to inferring the feedback after the initial training phase for a within-manifold perturbation, but it is definitely better compared to an outside-manifold perturbation (Fig 4D). Furthermore, the estimate is almost as good after 6 trials as after 50 trials (Fig 4D). These results suggest that feedback learning is possible for within-manifold perturbations, at least from a computational point of view, and that observing only a few trials is in principle enough to obtain a good estimate. In contrast, the feedback weights can not be learned for an outside-manifold perturbation.

To get intuition for this result, we firstly analysed a reduced model, where we synthesized data for any given Eigenvalue spectrum. For this reduced model, the output transformation only consists of the projection to the neural manifold, as this seemed to be the relevant part of the BCI mapping contrasting feedback learning for within- versus outside-manifold perturbations (S10 Fig). To learn feedback signals in this case, we regressed neural activity against 10-D neural mode activity. By measuring feedback learning performance for different Eigenvalue spectra, we found that high dimensional dynamics would allow feedback learning even for outside-manifold projections (S11 Fig). To understand this further, we returned to the simulation data and analysed feedback learning performance in more detail (S12 Fig). We used the fact that the feedback weight matrix can be described as two separate feedback vectors, one for the cursor's $x$-dimension and one for its $y$-dimension, each having the same number of entries as neurons in the network. Each entry in this vector is telling how neural activity of a specific neuron in the network should change in response to an observed error in cursor dynamics (either in $x$- or $y$-direction). Instead of using the basis system where each axis represents a neuron, we can instead look at these feedback vectors in the basis system given by principal component analysis (S12(C) Fig). In this view, each entry in the feedback vector tells how a neural mode should change according to an observed cursor error. As expected, for a within-manifold perturbation most of the entries of a feedback vector in this transformed coordinate

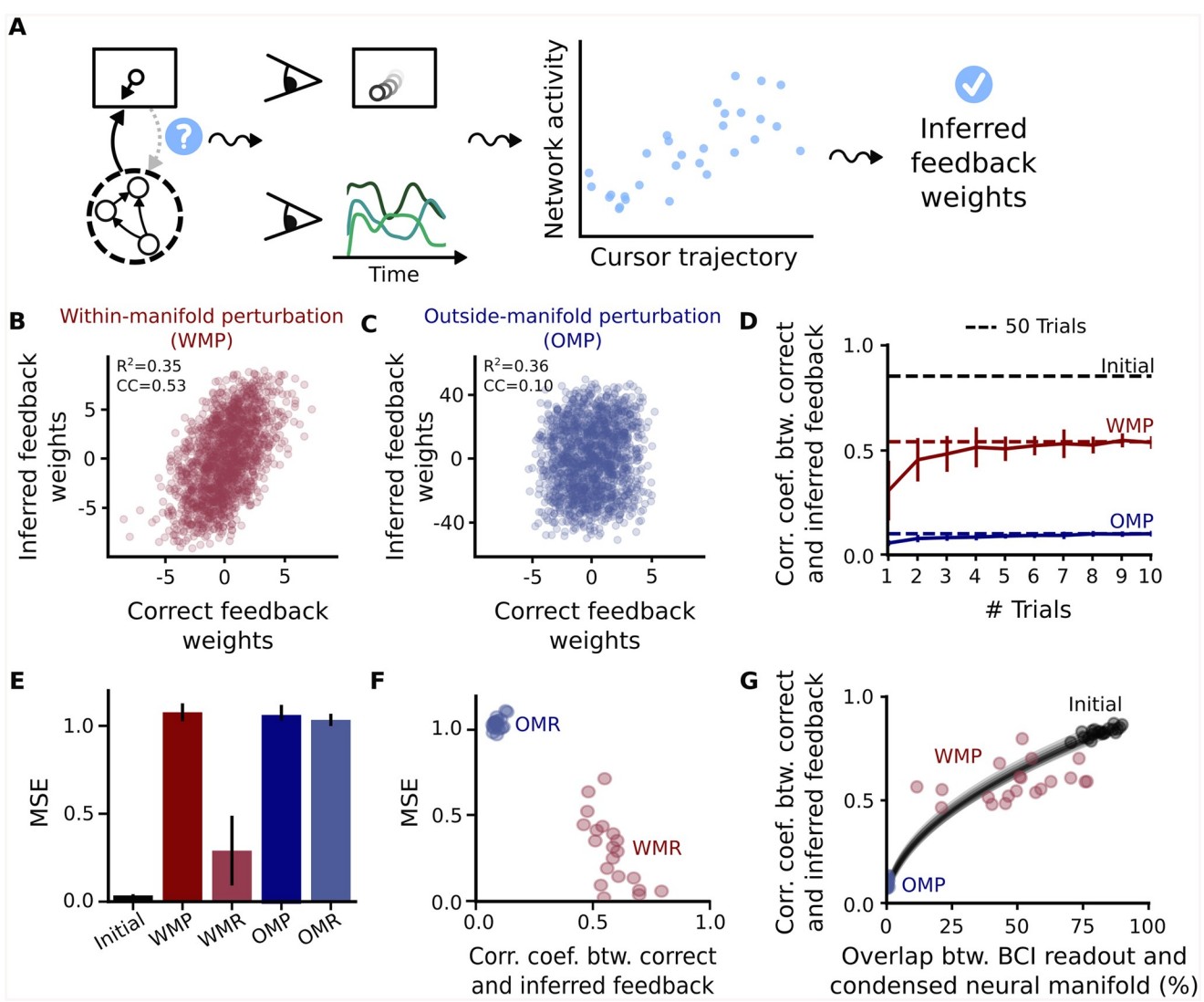

**Fig 4. Learning the feedback signal is possible for within- but not for outside-manifold perturbations.** (A) Regression is used to learn feedback weights. (B-C) Result of feedback learning for a within-manifold perturbation (WMP) (B) and an outside-manifold perturbation (OMP) (C). Accuracy of predicting neural dynamics from cursor dynamics is measured by $R^2$. Similarity between inferred and correct feedback weights is measured by calculating the correlation coefficient between both. (D) Feedback learning results depend on the number of trials used for regression. Dashed lines show the feedback learning result for taking 50 trials. Compared is feedback learning after the initial training phase (where BCI readout and internal manifold completely overlap) (black), after within-manifold perturbation (red) and after outside-manifold perturbation (blue). (E) Recurrent relearning using the learned feedback weights. As before, task performance is quantified by calculating the mean squared error (MSE) between target cursor velocity and produced cursor velocity. Compared is task performance after within- (WMP) or outside-manifold (OMP) perturbation to performance after relearning with inferred feedback weights (WMR and OMR respectively) (F) Accuracy in the feedback learning, measured by the correlation coefficient between correct and inferred feedback weights, affects recurrent relearning performance, measured by mean squared error. (G) Alignment of internal manifold and readout determines feedback learning performance. To take into account variance in feedback learning for within-manifold perturbations, we calculate the manifold overlap only up to a network-specific dimension, not up to 10 dimensions as done in the rest of the paper. The specific dimension is defined as $1 + \left(\sum \lambda_i\right)^2 / \sum \lambda_i^2$ where $\lambda_i$ are the Eigenvalues of the covariance matrix. The data points show feedback learning results for 20 simulations, whereas the black lines are interpolations between OMP and initial BCI mappings for which we evaluated feedback learning.

system are zero (S12(D) Fig). As the BCI readout only takes into account the first ten prominent neural modes, also the correct feedback vector is confined to these first ten modes (S12 (D), S12(F) and S12(G) Fig). In contrast, an outside-manifold perturbation results in a BCI readout, and thereby a feedback vector, which has components normally distributed across

principal component dimensions (S12(E) Fig). This means that basically every mode of the system has to adapt in order to counteract an outside-manifold perturbation. In this case it is not possible to infer the correct feedback weights for all of these modes (S12(F) Fig). Interestingly, we found that weights obtained from a linear regression are dominated by components of dominant neural modes (S12(H)–S12(J) Fig). This means that regressing neural dynamics against cursor dynamics does not put equal weight on predicting all neural modes, but instead focuses on predicting the dominant modes. That is also the reason why we observed equal accuracy in predicting neural dynamics from cursor dynamics under both a within-manifold perturbation and an outside-manifold perturbation (Fig 4B and 4C $R^2$ values). In both cases, feedback learning is able to predict the dominant modes with some accuracy, and to infer the corresponding feedback weight components (S12(G) Fig). For within-manifold perturbations, the feedback weight components of the prominent modes are the ones which solely constitute the feedback signal. The weight inference is therefore successful. In contrast, for the outside-manifold case, the feedback weight components of the dominant modes constitute only a minor fraction of all feedback weights which would need to be inferred (S12(F) Fig) Weight inference therefore fails in this case. Projecting the feedback weights into the principal component axis of neural dynamics allowed us to understand why feedback learning is possible for a within-manifold perturbation, but impossible for an outside-manifold perturbation.

In a next step, we wanted to test whether the learned feedback weights are good enough to correctly drive recurrent relearning. For this, we estimated the feedback weights offline as described above and then used them during recurrent relearning to propagate the error signal from cursor to neuron level (S12(C) Fig). As expected, we found that outside-manifold learning will be completely impaired if we use the learned feedback weights (Fig 4E blue). In contrast, task performance increased during recurrent relearning for within-manifold perturbations (Fig 4E red). Interestingly, we found that there is a relatively large amount of variation in relearning performance if we compare different network initializations. However, this large amount of variation is only observed for the task performance after within-manifold relearning. We hypothesized that the accuracy of the learned feedback weights is responsible for the successful retraining of recurrent weights during the adaptation phase. This is indeed what we found when we plotted the retraining performance against the feedback learning accuracy (Fig 4F). In summary, these results suggest that the accuracy of the feedback learning determines whether or not the BCI perturbation can be counteracted by recurrent restructuring. Together with the fact that it is not possible to infer feedback weights for outside-manifold perturbations, this could explain why outside-manifold learning was not observed experimentally [32].

Although we have shown that the accuracy of the feedback signal is responsible for either success or failure of recurrent retraining, it is still unclear why feedback learning works better for some BCI perturbations than others, and especially why it works for within-manifold perturbations, but completely fails for outside-manifold perturbations. As the main difference between within- and outside-manifold perturbations is the alignment of the readout to the internal manifold, we hypothesized that this alignment controls whether feedback learning is successful or not. To test this, we carried out simulations where we interpolated between the initial BCI readout and the one used for an outside-manifold perturbation. This allowed us to sweep the whole spectrum of alignments, as the initial BCI readout is completely aligned to the internal manifold of the network, whereas the readout for outside-manifold perturbations is completely orthogonal to it. We then inferred the feedback weights for all of these different readouts and quantified the result by calculating the correlation coefficient between the learned and the correct feedback weights. As hypothesized, we found that there is a specific monotonic relationship between feedback learning performance and alignment of BCI readout

and internal manifold (Fig 4G). This also explains why we observed a large variation for within-manifold relearning performance. In our setup, we fix the number of manifold dimensions used in the BCI readout, but for each individual network the internal manifold dimension can deviate from this. This means that within-manifold perturbations can be in fact slightly outside-manifold for individual networks, so that the BCI readout in these cases is not hundred percent aligned to the true internal manifold. In summary, by looking at the underlying requisites enabling successful feedback learning, we could understand why it is impossible to infer correct feedback weights in the case of an outside-manifold perturbation and why we would expect variations in within-manifold learning in the case that the readout dimension does not exactly match the internal manifold dimension.

Although we have shown so far that it is in principle possible to infer feedback weights, the question remains how this procedure could be implemented in the brain. Obviously, linear regression is a statistical method which is unlikely to be directly implemented by the brain. However, a biologically plausible learning rule can be used to learn to predict neural dynamics from cursor dynamics, and thereby successfully learn feedback weights (S17(A) and S17(B) Fig). Another major concern is the multiple use of the feedback weights. Firstly, they are learned using the neural dynamics, but then they are used to propagate the error signal. To tackle this problem we have alternatively tried to learn feedback signals not by regression neural dynamics against cursor dynamics, but instead by regressing neural dynamics directly against cursor errors. We found that it is possible to infer correct feedback weights in this scenario, as long as the produced cursor dynamics are larger than the target cursor dynamics (S17 (D)–S17(F) Fig). Thereby, we can imagine a neural implementation where neurons representing the error in cursor dynamics are connected to neurons in motor cortex, responsible for moving the cursor. The weight between them would then be adjusted over time to learn to signal the correct error feedback. An open question which remains is how neurons in motor cortex can simultaneously represent two signals, the error and the internal dynamics. Interestingly, this could be implemented by neural multiplexing [40], for example by dendritic and somatic signalling [41]. Taken together, we showed that our approach of learning feedback signals can be realised using local, biologically plausible, synaptic plasticity.

## Feedback signal for outside-manifold perturbation can potentially be learned with incremental strategy

Besides the original experimental study showing that monkeys can not adapt to outside-manifold perturbations on the timescale of a single experiment [32], there is new evidence showing that outside-manifold learning is indeed possible if the training lasts for several days and follows a specific incremental protocol [33]. We have shown so far that outside-manifold learning should not be possible, as the correct feedback weights needed for successful retraining can not be learned. We reasoned that to nevertheless be able to learn outside of the original manifold, one would have to insert intermediate perturbation steps, where for each perturbation step there is a certain amount of overlap between the current internal manifold and the BCI readout. This would assure that the system has the chance to learn constructive feedback weights in each step, which could then lead to successful recurrent restructuring. With this strategy, one could potentially build up the learning by shifting the internal manifold step-by-step closer to the one defined by the outside-manifold perturbation. To test this idea we implemented the incremental strategy by inserting three intermediate perturbations which interpolate between the original BCI readout and the readout for an outside-manifold perturbation (Fig 5A).

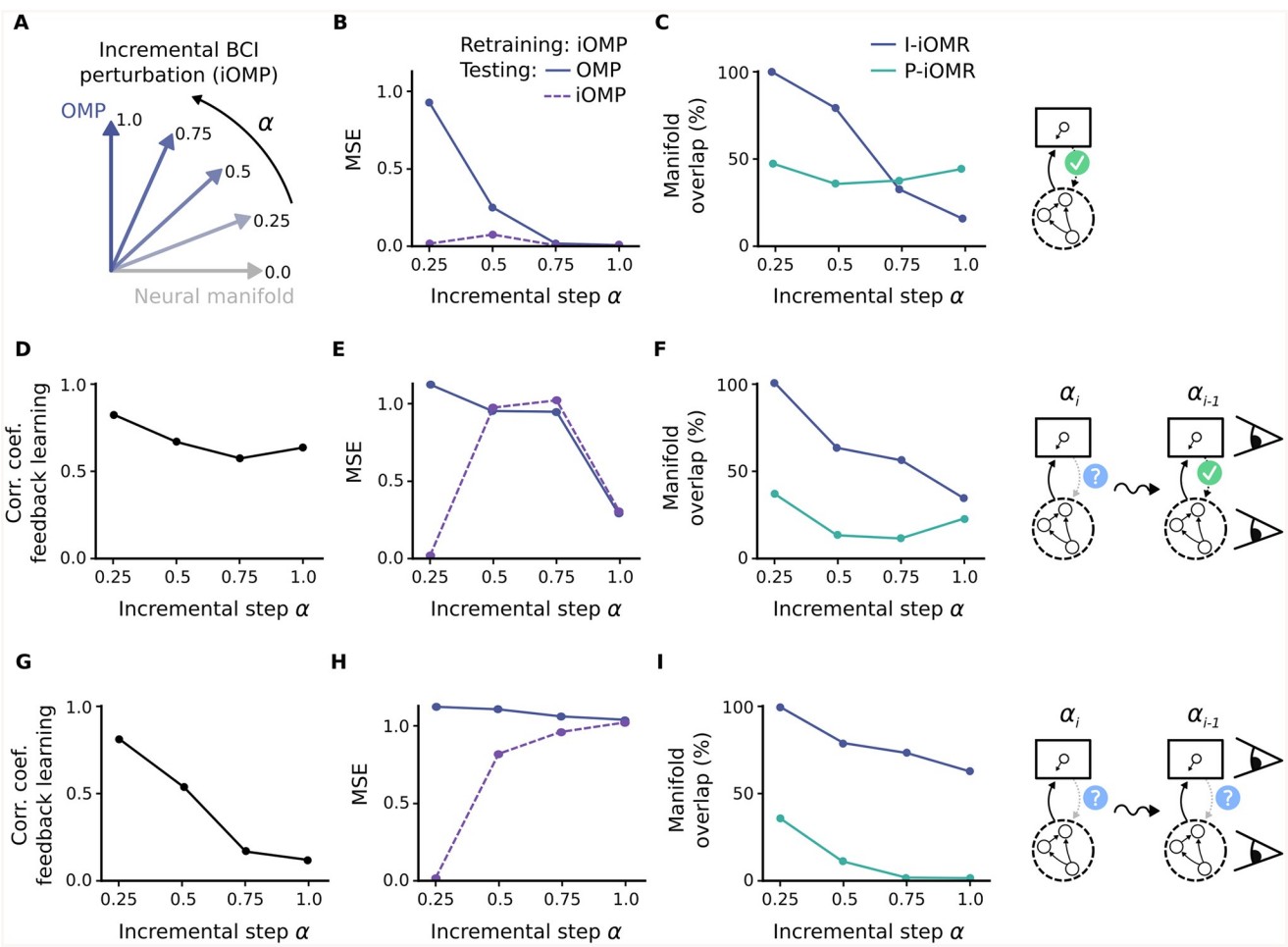

**Fig 5. Feedback signal for outside-manifold perturbation can potentially be learned with incremental strategy.** (A) Incremental outside-manifold perturbation, where with each incremental step $\alpha$ the BCI readout is less aligned with the original neural manifold. (B-C) Retraining results for incremental perturbation steps using correct feedback signal. (B) Task performance after retraining, measured as mean squared error (MSE), achieved with current incremental BCI readout (iOMP), which is the readout used during retraining, as well as achieved under the full outside-manifold perturbation (OMP). (C) Manifold overlap between initial manifold and manifold after relearning (blue line), as well as between target manifold, defined by the current BCI readout, and manifold after relearning (green line). (D-F) Retraining results using incremental strategy and assuming, that each consecutive incremental learning step starts from a correctly trained network. (D) Feedback learning performance, quantified by calculating the correlation coefficient between correct and inferred feedback weights, for each incremental learning step. (E) Task performance after recurrent retraining. (F) Manifold overlap for each incremental step. (G-I) Retraining results using full incremental strategy. Here, each consecutive learning step starts from the network state obtained in the previous step. Feedback learning performance (G), task performance after recurrent retraining (H) and manifold overlap (I).

To test whether the inclusion of the extra BCI perturbations is enough to enable feedback learning, we firstly performed a simplified simulation, where the feedback learning relies on a correctly trained previous network state (Fig 5D–5F). To obtain the correctly trained network state, we ran a separate retraining simulation with an ideal-observer feedback signal for every intermediate perturbation step. As expected, each of the incremental BCI perturbations could correctly be counteracted by recurrent retraining with an ideal-observer feedback signal (Fig 5B). By looking again at the manifold overlap obtained during retraining, we found that each retraining simulation achieved between 40%—50%, in line with our previous results (Fig 5C). To test feedback learning, we next tried to learn the correct feedback weights for each perturbation step by observing the correctly trained network from the preceding perturbation step. We found that the overlap between the network state of one perturbation step and the BCI

readout of the next step is high enough to allow successful feedback learning (Fig 5D). To finally test whether the incremental learning strategy would work in this simplified setup, we used the learned feedback weights to retrain the recurrent network. Interestingly, we found that retraining drastically improves by using an incremental training strategy (Fig 5E). However, the manifold overlap achieved during recurrent retraining is lower compared to retraining with an ideal-observer feedback signal (Fig 5E). This makes sense as the learned feedback signal does not completely point in the new desired directions, as it would do for the correct feedback signal. Instead, it also has components pointing in the dimensions of the manifold orientation before retraining. With this simplified scenario we could show that incremental learning can potentially be used to counteract an outside-manifold perturbation, assuming that the new incremental perturbation always starts from a well-trained network.

To finally see whether our network can be trained incrementally, without relying on correctly trained intermediate steps, we implemented a similar simulation as before, but now the feedback learning uses the preceding network states which also have been trained with learned feedback, which is not necessarily the correct feedback we have used before (Fig 5G–5I). We found that for this scenario the accuracy of the feedback learning drops for each intermediate perturbation step (Fig 5G). As expected, this deficiency led to poor performance in the recurrent retraining (Fig 5H) and the incremental strategy is therefore not helping in this case. To understand why incremental learning fails here, we compared what happens to the overlap between internal manifold and BCI readout with each consecutive step. In the simplified simulations before (Fig 5B–5F), we found that the internal manifold does not align as well with the current readout if we use learned feedback weights instead of the correct ones (compare Fig 5C and 5F). In the following incremental step, this reduced overlap leads to a larger misalignment between current network state and BCI readout, which in turn leads to a decrease in feedback learning accuracy. The problem is therefore that the network can not catch up with the BCI perturbation and falls behind more and more for every incremental learning step. In summary, we demonstrated that a progressive schedule could in principle lead to outside manifold learning, but we also identified limitations associated with this schedule.

## Discussion

To investigate which factors influence learning within and outside of the neural manifold, we used a recurrent neural network trained with a machine learning algorithm. Although this method of modelling neural dynamics lacks biological details, using RNNs has been surprisingly useful to understand neural phenomena [23, 42–45]. Using this approach, we could identify feedback learning as a potential bottleneck differentiating between learning within versus outside the original manifold. As feedback learning might not be the only relevant factor, we also tested the impact of further biological constraints. We found that neither separation of excitatory and inhibitory neural population (S5 Fig), nor local learning rule (S6 Fig) [46, 47], qualitatively change the fact that within- and outside-manifold perturbations can equally well be learned, given an ideal-observer feedback signal. We also took into account the amount of weight change produced in order to assess learnability of within- and outside-manifold perturbations. Of course, the solutions the recursive-least-squares learning algorithm found are not guaranteed to be accessible through a biological learning algorithm and should be considered as an upper bound on weight change, as RLS is known to produce relatively large weight changes in order to keep the observed error small. Investigating the weight change under a more biologically plausible learning algorithm (S6 Fig) showed indeed much smaller weight changes with almost equal learning performance. We also saw a minor difference between

within- and outside-manifold learning with respect to the learning timescale (S6(A) Fig) and the amount of weight change (S6(C) Fig), but not the relearning performance (S6(B) Fig). This could potentially also contribute to the experimental observation that within-manifold learning is easier than outside-manifold learning.

In principle, there are two alternative scenarios which would lead to altered network dynamics in motor cortex. Assuming motor cortex is highly controlled by other brain regions, such as for example the cerebellum and pre-motor cortex, the first scenario would be to alter the control signals coming from these regions, and to therefore trigger different activity patterns [35]. Nevertheless, how the inputs are learned remains unclear. The second scenario is rewiring in the local circuit. In this study we focused on the latter and tested to what extent it can explain experimental observations [32, 33]. We found that reorganization of local connectivity does not necessarily lead to a change in neural manifold. This shows that globally coordinated changes in neural activity can not only be realized by adaptation of input drive, but also via local rewiring driven by a learned feedback signal. Interestingly, our results on feedback learning are not necessarily constrained to the hypothesis of local plasticity, but could be also relevant for adapting neural activity by changes upstream. It remains to be investigated to what extent one can contrast both frameworks, given experimental data, as biology potentially uses multiple mechanisms in parallel.

Beside the main result of Sadtler et al., showing that within-manifold learning is easier than outside-manifold learning, they made further experimental observations which we tested in our model. Firstly, Sadtler et al. observed a differential after-effect for within- versus outside-manifold learning. For a within-manifold perturbation, the monkey had to relearn the original mapping in order to achieve high task performance. In the case of an outside-manifold perturbation however, after switching back to the original mapping the monkey was immediately able to perform the task with high accuracy. Our current modelling approach does not prevent learning in the case of an outside-manifold perturbation. There are still changes in neural dynamics under an outside-manifold perturbation, even if they do not improve task performance, as the feedback signal is misaligned (Fig 4C and 4E). However, it seems plausible that there would be another mechanism to prevent learning under invalid feedback in reality, similar to the complex interaction of feedforward and inverse model relevant in motor control [48]. Secondly, in a follow up study Golub et al. [49] discovered that the monkeys learned to counteract a within-manifold perturbation by reassociating existing neural activity patterns. By adapting their analysis we found that our simulation data for within-manifold learning was also best described by the reassociation hypothesis (S13 Fig). Thirdly, Oby et al. found that there are two different types of contributions to outside-manifold learning: 1) components which lie outside of the original set of activity patterns but still within the manifold and 2) true outside-manifold components which are orthogonal to the original manifold [33]. We found that our model too is using within-manifold components to learn outside-manifold perturbations (S14 Fig). Together, these results show qualitative agreement between our simulation data and a set of relevant experimental observations.

A recent computational study has shown that weight changes necessary for within-manifold learning are much smaller compared to outside-manifold learning and proposed this finding as a potential reason for the experimental observation in Sadtler et al. [34]. In contrast, we could not observe a difference in weight change during retraining (Fig 2C). However, we did observe that, if we constrain the number of plastic weights, within-manifold learning is more successful compared to outside-manifold learning (Fig 3C and 3D) and also that weight changes related to outside-manifold learning are slightly higher-dimensional (S4(C) and S4 (D) Fig). One fundamental difference compared to Waernberg and Kumar is that they used an alternative way of implementing the BCI perturbation, which led to the finding that the weight

changes are bigger during outside-manifold learning (S15 Fig). Instead of setting new target readout dynamics, which could be derived given a certain within- or outside-manifold perturbation, they implemented a perturbation by changing the encoder matrix (called $K$ in their case). As the encoder matrix injects the readout dynamics back into the system, this already changes the effective recurrent weights in the network, even before retraining. To counteract the perturbation, the readout weight matrix $\Phi$ is retrained to produce the same readout dynamics as initially. The overall weight change which Waernberg and Kumar reported in their paper is given by an outer product of both matrices $K\Phi$. By reimplementing their simulations, we replicated their finding that within-manifold learning leads to smaller weight changes than outside-manifold learning (S15(C), S15(E) and S15(G) Fig). Interestingly though, when we implemented the BCI perturbation by setting new target readout dynamics, instead of changing $K$ itself, the observed weight changes were small both for within- and for outside-manifold learning (S15(D), S15(F) and S15(H) Fig). Thereby, we could identify the main difference between both studies and replicate our finding that within- and outside-manifold learning produces similar amount of weight change if retraining starts from the initial connectivity, as it would be for a BCI monkey experiment.

As having a feedback signal is a general concept of motor control [48, 50, 51], it would be interesting to study whether one could also infer correct feedback signals in an upstream region exclusively for the case of a within-manifold perturbation. We hypothesize that, as long as the, potentially non-linear, output transformation via motor cortex is aligned with the internal manifold of the upstream region, feedback learning should be viable. Thus, our study gives a new perspective on the general problem of credit assignment, proposing that correct feedback learning critically depends on the alignment between output transformation and internal manifold in a specific neural pathway (Fig 4G).

Our results can also be related to a recent finding, reporting that a variety of different movements are controlled by neural activity lying in the same subspace across movements [29]. As our results suggest that the readout—used to produce a movement—indirectly defines the accessible neural activity space for motor learning (Fig 4F and 4G), we would expect that a stable readout implies a stable manifold across different movements. One could test this hypothesis by estimating the muscle output transformation for each specific movement. We would predict that less overlap in the output transformation implies less overlap in neural manifolds, measured during the two different movements.

Finally, our work might be relevant for future BCI designs. Our results support the idea that a subject can potentially learn to interact with any kind of static BCI mapping, as long as its dimensions are aligned to the internal dynamics of the local circuit it is connected to. It remains to be experimentally tested to what extent this holds true, but if so, it could impact development of brain-computer interfaces for real-world applications.

In summary, our model makes the following predictions, 1) for within-manifold perturbations, it is possible to infer useful feedback weights by regressing neural activity against cursor dynamics, 2) learning performance during a BCI task can be predicted by the alignment between BCI readout and internal manifold of the connected brain region, 3) silencing local network rewiring would impair learning of within- as well as outside-manifold perturbations.

In conclusion, we showed that local network rewiring can account for the observed behavioural differences between within- and outside-manifold learning, assuming that the feedback signal driving local reorganization has to be learned.

## Methods

### Task

In order to compare our learning results to experimental studies, we implemented a standard center-out-reach task, which is regularly used to experimentally probe motor learning. In our case, the task consists of six targets, evenly distributed on a circle. In an experiment, the subject would get a visual cue for example at the beginning of each trial to indicate which of the six targets is the target of the current trial. The subject should then reach this target in a given amount of time. We simulated the target cue by having an input pulse at the beginning of each trial, which is distinct for each of the six different targets. To simulate activity in motor cortex we used a recurrent neural network (RNN). The output is produced by a mapping, which we called brain-computer interface (BCI) for didactic reasons. It transformed the rates measured in the RNN to $x$ and $y$ velocity of a cursor, where the cursor simulated the center-out-reach movement. At the beginning of each trial, the cursor was set to the center of the workspace, and after the target cue ends, its movement was controlled by the RNN dynamics. The target speed for the cursor was 0.2 and the target directions were evenly distributed between 0 and $2\pi$. To measure performance we calculated the mean squared error between the target cursor velocity and the produced cursor velocity and summed it over all 50 test trials. As the target cursor velocity was constant and did not change depending on where the cursor is at a given moment, we used an open loop setup. The target for each specific trial was randomly chosen among the six.

### Recurrent neural network (RNN)

The recurrent neural network was simulated by the following dynamical equation:

$$\tau \dot{x}_i = -x_i + \sum_j W_{ij} r_j + s_{ik}$$

The recurrent connections $W_{ij}$ were sparse, with connection probability $p = 10\%$, and drawn from a Gaussian $\sim \mathcal{N}(0, g^2/(Np))$ (cf. list of parameters in Table 1). To simulate the target cue our network received a pulsed input $s_{ik}$ at the beginning of each trial. The input is given by

$$s_{ik} = W_{ik} \, s^{\text{pulse}}$$

where $s^{\text{pulse}}$ is a 0.2$s$ long pulse with amplitude 1 and the input weights $W_{ik}$ were drawn randomly from a uniform distribution $\sim \mathcal{U}(-1, 1)$.

**Table 1. List of parameters used in the model.**

| Variable | Definition | Value |
|---|---|---|
| $N_T$ | number of targets in center-out-reach task | 6 |
| $\tau$ | unit time constant | 0.1 s |
| $g$ | connection strength | 1.5 |
| $p$ | connection probability | 10% |
| $N$ | number of neurons in the network | 800 |
| $d_M$ | predefined manifold dimension | 10 |
| $L$ | trial duration | 2 s |
| $dt$ | time discretization | 0.01 s |

To implement learning in the network we used an adapted form of the recursive-least-squares algorithm, which was inspired from [18]. If learning is on, it started after the target cue period. Then, at every second time point (which corresponds to $\Delta t = 20ms$) an update step was made. Firstly, the performance error $e^P$, given by the difference between the produced cursor velocity and the target cursor velocity, was calculated. Via a linear transformation this two-dimensional performance error was then assigned to errors $e$ on the activity of single units in the network.

$$e = W^{fb} e^P$$

This transformation $W^{fb}$ is what we defined as feedback weights. In the ideal-observer case, this feedback matrix was given by the pseudo-inverse of the BCI transformation. Note that we calculated the error for each neuron through these feedback weights, which was used to adapt recurrent weights in the network, but there was no additional input for each neuron due to this error signal. The recurrent weight update is given by

$$W_{ij}(t) = W_{ij}(t - \Delta t) - \frac{e_i(t)\sum_k P^i_{jk}(t)r_k(t)}{1 + \sum_m \sum_n r_m(t)P^i_{mn}(t - \Delta t)r_n(t)}$$

where $e_i$ is the error for unit $i$ and $P^i$ is a matrix which estimates the inverse of the correlation matrix of the input to neuron $i$. $P^i$ was also updated in every update step and the update rule is given by

$$P^i_{jk}(t) = P^i_{jk}(t - \Delta t) - \frac{\sum_m \sum_n P^i_{jm}(t - \Delta t)r_m(t)r_n(t)P^i_{nk}(t - \Delta t)}{1 + \sum_m \sum_n r_m(t)P^i_{mn}(t - \Delta t)r_n(t)}$$

At the start of learning, the matrices $P^i$ were initialized as diagonal with 0.05 on the diagonal. For each training period the network was trained for 80 trials. Note that the training algorithm did not change the structural connectivity, so only connections which were non-zero at the beginning were updated.

## Brain-computer interface (BCI)

The brain-computer interface consisted of two linear transformations which sequentially transformed network rates to cursor velocities. The first transformation is given by a projection to principal components of the network dynamics. To calculate this projection, we measured the dynamics over 50 trials during which the network is performing the task. Here, and at every other point, we took network dynamics into account only from the end of the cue phase. Then, we used principal component analysis to calculate the projection matrix $C$. The second transformation is given by a decoder matrix $D$ which was optimized offline by doing regression of target cursor velocities against the first 10 principal components, taking the same 50 trials as before. The full BCI transformation $T$ is then given by

$$T = D\ C$$

For the initial training period, $T$ was set up randomly. Each entry of the matrix was drawn from a standard normal distribution, normalized to have matrix norm 1 and multiplied with a factor 0.04. This setup showed to drive initial learning sufficiently well, and assured good task performance after the initial network training period, followed by the BCI tuning described above (S2 Fig). Note that the initial random mapping had high impact on the produced neural manifold and therefore also on the optimized BCI mapping (S9 Fig).

For a within-manifold perturbation, we inserted a permutation matrix $\eta_{WM}$ in between the first and the second linear transformation, which shuffled the decoder weights related to the principal components.

$$T_{WM} = D \ \eta_{WM} \ C$$

In contrast, for an outside-manifold perturbation, we inserted a permutation matrix $\eta_{OM}$ before the first linear transformation, which shuffled the projection matrix and therefore resulted in not extracting the correct principal components of the network dynamics.

$$T_{OM} = D \ C \ \eta_{OM}$$

In order to assure that both types of perturbations cause approximately the same performance error we calculated the performance, measured as mean squared error, for 200 random perturbations of each type. Then we calculated the mean performance over all perturbations, ranked them according to their difference to the mean and selected the perturbations closest to the mean. This ensured that the selected perturbations had similar mean squared error. Note that although the performance error was matched through this procedure, the cursor velocities for outside-manifold perturbations were scaled down compared to within-manifold perturbations. This is a consequence of the readout transformation to non-dominant directions in neural state space which naturally contain less variance and therefore lead to lower cursor velocities.

## Analysis

To quantify manifold overlap we adapted the measurement used in [49]. Firstly, we calculated the covariance matrix of the network dynamics $S$, given 50 trials of performing the task. This was done for the initial network (subscript $_1$) and for the network after retraining (subscript $_2$). Then, both of these matrices were projected onto the neural manifold defined by the initial network. This projection is given by the same projection matrix $C$ as used in the BCI transformation. To quantify the amount of explained variance in the first 10 dimensions, defined as $\beta_1$ for the initial network and $\beta_2$ for the retrained network, we calculated the trace of the projected matrices (taking into account only the first 10 dimensions) and divided it by the trace of the original matrices (full-dimensional).

$$\beta_2 = \frac{Tr(CS_2C^T)}{Tr(S_2)} \ \ \beta_1 = \frac{Tr(CS_1C^T)}{Tr(S_1)}$$

The manifold overlap is then defined as the ratio $\beta_2/\beta_1$. For calculating the overlap between the perturbed BCI readout and the network manifold after retraining we replaced the projection into the Eigenbasis of the initial covariance matrix, given by $C$, with a projection given by the new readout transformation, which is $C \eta_{OM}$.

To calculate the overlap between BCI readout and condensed network manifold (Fig 4G), we summed up the normalized projections of the readout $T$ into the first $k$ dimensions of the Eigenspace of the network dynamics, given by $C$,

$$\frac{\sum_i^k (T_xC_i)^2}{2|T_x|^2} + \frac{\sum_i^k (T_yC_i)^2}{2|T_y|^2}$$

where $T_x$ is the first row of the BCI transformation, decoding the cursor velocity in x-direction, and $T_y$ is the second row of the BCI transformation, decoding the cursor velocity in y-direction.

The number of dimensions $k$ is defined by

$$1 + \frac{\left(\sum_i \lambda_i\right)^2}{\sum_i \lambda_i^2} = 1 + PR$$

where $\lambda_i$ are the Eigenvalues of $S_1$ and $PR$ is participation ratio, a commonly used measure of dimensionality.

### Feedback learning (Fig 4)

To learn the feedback weights for a given BCI perturbation, we created a dataset of 50 trials where the network is performing the task with the perturbed BCI readout. During this, there are no plastic changes yet in the recurrent connections. To infer feedback weights we then regressed the network rates, measured during the 50 trials, against the x and y cursor velocities. The accuracy of the regression is about the same for within- and outside-manifold perturbations, whereas the match between inferred and true feedback weights strongly differs. To obtain Fig 4D, we only used a randomly selected subset of the 50 trials. To obtain Fig 4G, we analysed not only the given BCI perturbations, but added intermediate perturbations, which interpolate between the original BCI transformations and the outside manifold perturbations. The analysis of the interpolated perturbations was shown in Fig 4G (black lines).

### Incremental strategy (Fig 5)

In each incremental step, the learning starts from the weight configuration after initial training. We did this to insure that the network stays in a physiological state, as the recursive-least-squares algorithm tends to increase the weights during training. Without this reset, the weight distribution would grow with every incremental training and could potentially lead to a network state where most of the neurons are saturated.

Each incremental BCI perturbation $T_\alpha$ is defined by interpolating between the original readout $T$ and the full outside-manifold perturbation $T_{OM}$, where the incremental factor $\alpha$ defines the ratio between both.

$$T_\alpha = (1 - \alpha)\ T + \alpha\ T_{OM}$$

### Supporting information

**S1 Fig. Control simulations.** (A) Original setup as in Fig 2. Upper left panel shows task performance, measured as mean squared error (MSE), after initial training (O), after within-manifold perturbation (WP), after within-manifold retraining (WR), after outside-manifold perturbation (OP) and after outside-manifold retraining (OR). Upper right panel shows the standard deviation of the weight change distribution between before and after retraining for within- (W) and outside-manifold (O) retraining. Lower left panel shows the manifold overlap between initial and retrained manifold for within- (W-o) and outside-manifold (O-o) perturbation, as well as the overlap between retrained and target manifold for outside-manifold perturbations (O-p). Lower right panel shows the mean principal angle between the same manifold as in lower left panel. (B) Position decoding instead of velocity decoding. (C) BCI perturbations are scaled in order to preserve the range of velocities after perturbation. This especially affects outside-manifold perturbations as in this case the velocity values are normally only 10% of the original ones, due to a signal loss caused by the outside-manifold projection. (D) BCI perturbations are scaled to assure that the theoretical approximation for the neural activity after retraining, given the initial training state, does not exceed the dynamic range of

the neurons. (E) Instead of having one input unit for each target, here, there are only two input units. They signal x and y position of the targets. (F) Reduced target speed of 0.01, instead of 0.2.
(PNG)

**S2 Fig. Details of initial network training.** (A-D) Initial training with random decoder, normalized and scaled with factor 0.04 (the default version used in the paper). (E-H) Initial training with random decoder, normalized and scaled by a factor 0.2. (A) and (E) Cursor velocities before and after training. (B) and (F) Eigenvalue spectrum of network dynamics before and after training. (C) and (G) Rate distribution before and after training. (D) and (H) Reconstructed cursor trajectory after training.
(PNG)

**S3 Fig. Simulation results for recurrent neural network trained with backpropagation-through-time using Pytorch.** Here, the network dynamics are given by $h_t = \tanh(W_{ih} x_t + W_{hh} h_{(h-1)})$. Each trial has 100 time steps and the target cue is given during the first 5 steps. The input $x_t$ is modelled similar to our main setup. The stimulus amplitude is 50. The incoming weights $W_{ih}$ are fixed and randomly drawn from a uniform distribution between $-\sqrt{1/N}$ and $\sqrt{1/N}$, where $N$ is the number of neurons in the network. The recurrent weight matrix is initialized in a similar fashion as in our main setup, except that here, we use a fully connected network. For gradient descent we use Adam optimizer [52] with learning rate 0.001. The loss and the fixed output decoder is the same as in our main setup. The training uses an ideal-observer feedback signal to propagate the error in cursor velocities to errors on single neurons in the recurrent network. For training, we use a batch size of 30. (A) Cursor velocities before initial training. (B) Cursor velocities after initial training. (C) Learning curve for initial training. (D) Performance results for within- and outside-manifold retraining. Number of training epochs for initial training, as well as retraining, is 1000. The MSE value in (C-D) is not summed over time or trials.
(PNG)

**S4 Fig. Weight change during retraining.** (A) Relation between weight before retraining and after retraining for within-manifold perturbation. (B) Relation between weight before retraining and after retraining for outside-manifold perturbation. (C) Measurement of the effective rank [53] of the weight change matrix for within-manifold and outside-manifold retraining. Both values are compared to values obtained for random rank-1, random rank-2 and full-rank matrices, having the same sparsity as the network weight matrix. (D) Dimensionality of weight change dynamics during retraining (dW measured at the end of each trial). Since it is not feasible to calculate the full covariance matrix for all plastic connections, as the number is too high ($\sim 64000$), we calculated the dimensionality from randomly chosen subsets of connections and checked for convergence.
(PNG)

**S5 Fig. Simulation with E-I network.** Simulation results for a network with Daleian connectivity. Each neuron has either positive (excitatory) or negative (inhibitory) outgoing weights. During learning, the sign of these weights is preserved. If a learning step would produce a sign flip we instead clip the weight to zero. (A) Training performance measured as mean squared error (MSE). (B) Standard deviation of weight change during within- and outside-manifold training. (C) and (D) Weights before versus after within- (C) or outside- (D) manifold training.
(PNG)

**S6 Fig. Simulation with local learning rule.** Implementation of recently proposed local learning rule approximating backpropagation-through-time algorithm [47]. The weight update is given by $dW_{ij}^t = -e_j^t(1 - \tanh^2(x_j^t))\sum_{t' \leq t-1} r_i^{t'}$, where $e_j$ is the error for neuron $j$, $x_j$ is the activity of neuron $j$ and $r_i = \tanh(x_i)$ is the rate of neuron $i$. (A) Learning curves for initial, within- and outside-manifold training period. Here, the mean squared error (MSE) is summed over update steps per trial, which are 90 steps. (B) Performance results after training measured as MSE. (C) Standard deviation of weight change during within- and outside-manifold training. (PNG)

**S7 Fig. Mode correlation between before and after retraining.** (A) Correlation between the neural modes before and after retraining for each of the six targets. Note, the neural manifold is in this case not the actual internal manifold (which is different after retraining), but the static one defined by the initial BCI mapping. We averaged over 20 simulations and the 6 targets. (B) Average mode correlation. Error bars are standard deviation across the ten modes. (PNG)

**S8 Fig. Mode alignment during outside-manifold and within-manifold relearning.** (A) Mode match is defined as scalar product between the Eigenvector of the specific Eigenvalue before and after retraining. Each dot represents the mode match of one of ten Eigenvalues considered, and shows the result for one of 20 simulation runs (total 200 points shown). We defined a mode as matched if the scalar product is bigger than 0.2. (B) We quantified how many modes are matched per simulation run, taking the definition from (A). (C) Same as in (A), but now for within-manifold retraining. As the decoding weights are shuffled during a within-manifold perturbation one can compare the mode match to the decoding weights before and after perturbation. Here, we compared to before perturbation. (D) Mode match for within-manifold retraining compared to decoding weights after perturbation. (PNG)

**S9 Fig. Initial random decoder shapes neural manifold and final BCI mapping.** (A-B) Standard case where initial random decoder output is two-dimensional. (A) Mode match is calculated as scalar product between vectors. The initial decoder induces the first two strong neural modes, which are strongly amplified in the Eigenvalue spectrum after initial training (B). (C-D) Test case where initial random decoder output is four-dimensional. Here, the assignment of initial decoder and neural manifold modes is less prominent, meaning that there is no one mode specifically responsible for one output dimension. (C). In this case the number of amplified modes does not correspond to the dimensionality of the initial decoder (D) (E) Initial random decoder shapes Eigenvalue spectrum during initial network training phase, and Eigenvalue spectrum in turn shapes optimized BCI decoding weights $D$. (F) Relation between optimized BCI decoder and initial random one used for initial network training. (PNG)

**S10 Fig. Feedback learning on mode level.** (A-B) Mode activation during one example trial for a within-manifold perturbation (A) and an outside-manifold perturbation (B). Mode activation is obtained by projecting measured neural dynamics onto the original neural manifold, given by the transformation matrix $C$. (C-D) Results of linear regression to infer inverse of transformation matrix $C$ for a within-manifold perturbation (C) and an outside-manifold perturbation (D). (PNG)

**S11 Fig. Broader Eigenvalue spectrum improves feedback learning performance for outside-manifold perturbations.** (A) and (C) Imposed Eigenvalue spectra, defined by $f(x;\gamma) =$

$\exp^{-x/\gamma}$. Dimensionality is calculated by $(\sum_i \lambda_i)^2 / \sum_i (\lambda_i^2)$ where $\lambda_i$ is the $i$th Eigenvalue. Detailed methods are described in S1 Appendix (B) and (D) Feedback learning results measured by correlation coefficient between inferred and true feedback weights, dependent on the dimensionality of the imposed Eigenvalue spectrum.
(PNG)

**S12 Fig. Inferring feedback weights is biased towards dominant neural modes.** (A-B) Feedback learning results for within-(WMP) (A) and outside-manifold perturbation (OMP) (B). CC describes the correlation coefficient between inferred and correct feedback weights. Same data as in Fig 4B and 4C. (C) Feedback weight vectors for x- and y-component of cursor dynamics can be expressed in terms of principal components of neural dynamics. With this, we can compare feedback learning performance, dependent on the related neural mode. Per definition, the most prominent neural modes in the system lie in the first principal component axes. (D-E) Distribution of projections into the principal component basis for correct feedback (x-component) for within-manifold perturbation (D) and outside-manifold perturbation (E). (F) Comparison of projections into the principal component basis for correct and learned feedback vectors (x-component). (G) Same as (F) but zoomed in to show only the first 20 principal components. (H-J) Absolute value of projection into principal component basis for inferred and correct feedback weights for within- (H) and outside-manifold perturbation (J). The inlets show the same plot as (H-J), but focussing on principal components 1-20.
(PNG)

**S13 Fig. Simulation results for within-manifold learning are in line with reassociation hypothesis [49].** (A) Cursor velocities under original (upper row) and perturbed mapping (lower row) for original (black) and retrained network (red). Comparison between simulation data (first column), realignment hypothesis (second column), rescaling hypothesis (third column) and reassociation hypothesis (fourth column) [49]. Detailed methods are described in S2 Appendix. B) Original neural activity (left), neural activity after within-manifold relearning (middle) and neural activity after outside-manifold relearning (right) (cf. Fig 2). Neural activity is projected into the original neural manifold and the first two principal components are shown. Different colors correspond to different target locations. Dashed black circle represents boundary of initial repertoire. (C) Change in covariability along original and perturbed mapping. (D) Change in variability of neural modes dependent on the imposed change in the BCI readout (pushing magnitude).
(PNG)

**S14 Fig. Within-manifold contributions to incremental outside-manifold learning [33].** (A) Relearning performance under incremental strategy (Fig 5E). (B) Cursor velocities under the original mapping for each incremental training step. Dashed black circle shows boundary of initial repertoire. Colours represent different targets. (C) Percentage of within-manifold patterns which lie outside of the original repertoire (dashed black circle in (B)). (D) Percentage of variability measured outside of original manifold as a function of incremental step $\alpha$. (E) Histogram of percentage of within- versus outside- manifold contributions. For each relearned neural activity pattern which was outside of the original manifold or repertoire we calculated the projection onto the original manifold $d_{WM}$ (within-manifold contribution) and subtracted this projection from the total neural activity vector to obtain the residual vector $d_{OM}$ (outside-manifold contribution). This gave two distances in the high dimensional space of neural activity. To calculate the percentage of within-manifold contribution we calculated $d_{WM}^2 / (d_{WM}^2 + d_{OM}^2)$.
(PNG)

**S15 Fig. Difference to previous study investigating the experimental results of Sadtler et al.**
**[34].** (A) Reimplementation of Waernberg and Kumar [34]. Neural modes are decoded from a
random recurrent network with static weights. The neural mode signal is fed back into the
recurrent network via a feedback matrix $K$. Network training consists of finding the decoding
weights $\phi$ which produce a given target signal at the neural mode level. FORCE is used to
adapt the decoding weights accordingly [17]. (B) Resulting neural mode dynamics after initial
training. (C) To implement the BCI experiment from Sadtler et al. Waernberg and Kumar
manipulated the feedback matrix $K$ and retrained the decoding weights $\phi$ in order to produce
the original target dynamics on the mode level [34]. (D) Alternative implementation of the
BCI perturbation which is closer to our simulations. Here, the BCI perturbation is imple-
mented by setting a new target signal on the mode level, without manipulating $K$. (E) Weight
change after perturbation, without retraining, for implementation of Waernberg and Kumar.
(F) Weight change after perturbation, without retraining, for alternative implementation. (G)
Weight change after retraining for implementation of Waernberg and Kumar. (H) Weight
change after retraining for alternative implementation.
(PNG)

**S16 Fig. Timecourse of relearning is similar for within- and outside-manifold perturbation**
**under the recursive-least-squares learning rule.** (A) Mean squared error (MSE) during
relearning for within- and outside-manifold perturbation. The MSE is summed over all update
steps in a trial, which constitutes 90 steps. (B) Amount of weight change measured during
relearning for within- and outside-manifold perturbation.
(PNG)

**S17 Fig. Learning feedback weights via a biological learning rule.** (A) Inferred feedback
weights using linear regression (LR) are compared to correct feedback weights, which are
given by the pseudo-inverse of the BCI mapping. In linear regression, weight factors are esti-
mated to predict neural dynamics from cursor dynamics. (B) Linear regression can be imple-
mented by a simple, biologically plausible, learning rule (BL). Learned feedback weights,
using a biological learning rule, are compared to correct feedback weights. (C) Illustration of
relearning steps: 1) Feedback weights are inferred from observing neural and cursor dynamics.
2) The learned feedback weights ($W^{fb}$) are then used to propagate the error form cursor to net-
work level. (D) Alternative way of inferring feedback weights. Instead of predicting neural
dynamics from cursor dynamics, neural dynamics could also be predicted by the observed
error signal in cursor dynamics. (E) Cursor dynamics under within-manifold perturbation are
in general smaller than the target values. Rescaling target values by 0.25 (dashed line) reverses
the relation. (F) Feedback learning performance, measured as correlation coefficient between
correct and inferred feedback weights, when cursor error signal is used to predict neural
dynamics.
(PNG)

**S18 Fig. Relearning under noisy feedback signal using recursive-least-squares algorithm**
**and local learning algorithm [47].** (A-B) Relearning performance, measured as mean squared
error (MSE), as a function of the amplitude of the noise in the feedback signal using recursive-
least-squares (RLS) algorithm (A) and an alternative implementation with a local learning
algorithm (Eprop) (B). (C-D) Manifold overlap between original manifold and manifold after
within learning (I-WMR), original manifold and manifold after outside learning (I-OMR) and
manifold defined by BCI perturbation and manifold after outside learning (P-OMR). (E-F)

Weight change during relearning. (A,C,E) Simulations with RLS (same as Fig 3B in the main paper). (B,D,F) Simulations with Eprop learning rule.
(PNG)

**S19 Fig. Simulation results are robust to how inputs are modelled.** (A-B) Upper left panel shows task performance, measured as mean squared error (MSE), after original training (O), after within-manifold perturbation (WP), after within-manifold retraining (WR), after outside-manifold perturbation (OP) and after outside-manifold retraining (OR). Upper right panel shows the standard deviation of the weight change distribution between before and after retraining for within- (W) and outside-manifold (O) retraining. Lower left panel shows the manifold overlap between original and retrained manifold for within- (W-o) and outside-manifold (O-o) perturbation, as well as the overlap between retrained and target manifold for outside-manifold perturbations (O-p). Lower right panel shows the mean principal angle between the same manifold as in lower left panel. (A) Result for simulations where inputs are modelled as a pulse (same as S1A Fig). (B) Results for simulations where inputs are modelled as tonic input. (C-D) Reconstructed cursor trajectories for pulsed input simulations (C) and tonic input simulations (D).
(PNG)

**S20 Fig. Simulation results are robust to noise.** (A-B) Upper left panel shows task performance, measured as mean squared error (MSE), after original training (O), after within-manifold perturbation (WP), after within-manifold retraining (WR), after outside-manifold perturbation (OP) and after outside-manifold retraining (OR). Upper right panel shows the standard deviation of the weight change distribution between before and after retraining for within- (W) and outside-manifold (O) retraining. Lower left panel shows the manifold overlap between original and retrained manifold for within- (W-o) and outside-manifold (O-o) perturbation, as well as the overlap between retrained and target manifold for outside-manifold perturbations (O-p). Lower right panel shows the mean principal angle between the same manifold as in lower left panel. (A) Standard simulations where no noise is injected (same as S1A Fig). (B) At each time step each neuron receives an extra input, drawn from a normal distribution with zero mean and standard deviation $f$. The noise for each neuron is independent. (C) Reconstructed trajectories for simulations without noise. (D) Reconstructed trajectories when noise (f = 1) is injected. (E-F) Network cannot be trained to perform the task if noise is too high (f = 10).
(PNG)

**S1 Appendix. Simplified model to investigate feedback learning.**
(PDF)

**S2 Appendix. Adaptation of analysis performed in Golub et al. 2018.**
(PDF)

## Acknowledgments

We thank A. Kumar, J.A. Hennig, S.M. Chase, B.M. Yu, A.P. Batista for discussions and J.A. Gallego and E.R. Oby for discussions and comments on the manuscript.

## Author Contributions

**Conceptualization:** Barbara Feulner, Claudia Clopath.

**Data curation:** Barbara Feulner.

**Formal analysis:** Barbara Feulner.

**Funding acquisition:** Claudia Clopath.

**Investigation:** Barbara Feulner, Claudia Clopath.

**Methodology:** Barbara Feulner, Claudia Clopath.

**Project administration:** Claudia Clopath.

**Resources:** Claudia Clopath.

**Software:** Barbara Feulner.

**Supervision:** Claudia Clopath.

**Validation:** Barbara Feulner.

**Visualization:** Barbara Feulner.

**Writing – original draft:** Barbara Feulner.

**Writing – review & editing:** Claudia Clopath.

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
