## [Decision Letter · Decision Letter 0]

8 Sep 2020

Dear Dr. Clopath,

Thank you very much for submitting your manuscript "Neural manifold under plasticity in a goal driven learning behaviour" for consideration at PLOS Computational Biology.

As with all papers reviewed by the journal, your manuscript was reviewed by members of the editorial board and by several independent reviewers. In light of the reviews (below this email), we would like to invite the resubmission of a significantly-revised version that takes into account the reviewers' comments.

The reviewers give a number of constructive suggestions on how to better support the conclusions drawn in the paper, and also point out where discrepancies with earlier research on this topic need to be addressed and clearly elucidated.

We cannot make any decision about publication until we have seen the revised manuscript and your response to the reviewers' comments. Your revised manuscript is also likely to be sent to reviewers for further evaluation.

Sincerely,

Abigail Morrison

Associate Editor

PLOS Computational Biology

Lyle Graham

Deputy Editor

PLOS Computational Biology

Reviewer's Responses to Questions

**Comments to the Authors:**

Reviewer #1: Summary

In this manuscript, the authors use a recurrent neural network (RNN) modeling approach to investigate the origins of the different types of learning observed in a series of recent BCI studies. The key experimental result from these prior studies is that neural activity decoded from primary motor cortex of behaving monkeys can be adjusted rapidly (i.e., within a single behavioral session) following a BCI perturbation if the new desired patterns of activity respect the pre-existing covariations within the decoded neural population (so-called within-manifold perturbation, or WMP). If not (i.e., in the case of outside-manifold perturbation, OMP), learning is much slower and requires an iterative training approach that tries to minimize violations in the natural covariation structure of neural activity at every step of the procedure.

The present study is potentially appealing in that it provides one explanation for why this dichotomy might exist. Using an RNN model trained on the same task used in the origin BCI study, the authors explore the hypothesis that learning performance may depend on the nature of the synaptic weight changes required by WMP versus OMP. First, they show that both types of perturbations could in theory be learned equally well in the ideal case where weights are allowed to change directly based on feedback in order to maximize performance. Second, they assess how WMP and OMP can be learned in more biologically-plausible conditions, i.e., with imperfect feedback or sparse plasticity. Under these conditions, WMP appears to be generally more learnable than OMP. Third, they test how well the model can infer feedback signals from its own internal dynamics. Again, in this case, WMP appears to outperform OMP. Finally, they show that OMP learning can be improved if the model is trained in a stepwise manner reminiscent of what was done in the original BCI studies.

Overall impression

The manuscript is written clearly, and most claims are appropriately supported by modelling data. The authors make a reasonable case that, should learning indeed engage plasticity mechanisms within M1, the stark differences between WMP and OMP learning observed in BCI studies could in principle result from imperfect feedback information driving incomplete plastic changes. One potential concern is that in their investigation, the authors completely overlook the possibility that learning may occur upstream of the decoded neural population, and hence may not require any recurrent connectivity changes within M1, particularly for WMP. Although the authors do mention this possibility in the discussion section, the fact that it is not at all explored in the model slightly undermines the impact of the study. Moreover, it is unclear how the results favoring WMP versus OMP learning can be explained by the learning strategy used to update RNN weights during retraining. Indeed, because the feedback learning is based on regressing the network internal dynamics against cursor movements, it seems that WMP may be trivially at an advantage. Finally, it remains unclear how the different types of feedback signals, even in their most sophisticated forms considered at the end of manuscript, could indeed operate inside the brain. Nevertheless, assuming that BCI learning is indeed related to connectivity changes, not to upstream inputs, and driven by feedback signals that can somehow be inferred from behavior, this study provides a characterization of this specific learning regime.

Major comments

Below are general comments that we ask authors to carefully consider and implement in the revised version of the manuscript.

1) As mentioned above, it is not clear why learning should necessarily depend on recurrent changes within the decoded populations of neurons (or in this case, the RNN connectivity). Indeed, particularly for the case of WMP, changes in inputs may very well be able to drive the network state to the desired part of the pre-existing neural manifold to accommodate the BCI perturbation. In fact, following the original BCI study [Sadtler et al, 2014], the same group later reported [Golub et al, 2018] that animals adapt to WMP through a ‘neural reassociation’ strategy whereby pre-existing activity patterns are recycled and reassigned to new intended cursor movements. The authors of the present study do not currently discuss whether the mechanism they propose for WMP (through weight changes without any change in inputs) could lead to the same observations.

2) Another important point missing from the discussion is the fact that in the most recent BCI study cited [Oby et al, 2019], it was found that OMP learning was not simply the result of outside-manifold changes. A large fraction of the changes was in fact within-manifold (but in such a way that projections on the not-completely-orthogonal readout dimension could be adjusted). This result seems to be loosely consistent with the findings of the current study, and the authors should discuss whether their model predicts (or could accommodate) this type of solution for OMP learning.

3) In the first part of the manuscript, where weight changes under ideal feedback are compared for WMP versus OMP, the authors make the point that weight change distributions are similar between the two perturbations. However, later in the manuscript, it is also stated that the dimensionality of these weight changes differs quite a bit between WMP and OMP (WMP leading to lower dimensional weight changes that OMP; Figure S4). Could that feature by itself already explain the difference in learning timescales between the two perturbations? Presumably, if weight changes are more correlated in WMP, they may occur more rapidly/easily, and may therefore already put WMP and OMP on an unequal footing even in the case of ideal feedback.

4) Currently, there is no consideration of learning timescale, only learning feasibility, between WMP and OMP. That is, the model is judged based on its capacity to produce the desired output post retraining. Because the original BCI findings pointed to differences in learning feasibility as well as timescales between WMP and OMP, it seems relevant to assess the model in terms of timescale too. Concretely, do the number of trials needed to reach max performance for WMP systematically differ from OMP? This will most likely depend on the learning rule used (as seen in Figure S6), but it would be worth providing some data on this.

5) One feature the authors do not currently rely on to constrain and further test their model is whether WMP or OMP learning would lead to an after-effect once the perturbation is removed. In the original BCI study, an after-effect was only seen after WMP, but not after OMP, presumably because no learning had occurred for this perturbation. Again, this seems consistent with what the authors of the present study claim (the fact that inferring feedback weights is only possible for WMP) and should be briefly mentioned in the discussion section.

6) It is stated that the initial performance decrements following WMP or OMP are similar (Figure 2B). Similarly, in the method section related to Figure 4, it is stated that the accuracy of the regression between network rates and x and y cursor velocities were “about the same” for WMP and OMP. How can this be? Since following an OMP, neural activity should have a smaller projection onto the new readout, shouldn’t we expect an OMP to lead to minimal cursor movement after perturbation? In the method section, it is said that OMP perturbations are selected to achieve the same performance decrements as in WMP; doesn’t that affect what types of OMP are indeed applied? I’m concerned that this biased procedure may generate OMP that no longer fit the features of OMP experienced by monkeys in the original BCI studies...

7) How does the regression procedure relating network dynamics to cursor movements work exactly? If the feedback learning is based on regressing activity against cursor movements, what this procedure gives is a set of target activities for every neuron. The authors should explain with an appropriate level of detail how these target activities are then turned into weight changes. It is not immediately obvious how the recursive-least-squares algorithm used in the ideal-observer case should be extended to work in the regression-based learning case. Moreover, it is unclear whether this procedure puts WMP at an advantage over OMP during retraining. If the regression is based on activity patterns that lie within a pre-existing manifold, wouldn’t the regression be able to extrapolate and predict new cursor movements better within the manifold than outside?

8) As I understand it, the rationale for adding noise on the ideal feedback signal, or making it partially available to the network, is to move away from biological implausibility. However, it is unclear how these conditions are in fact more realistic than the ideal case itself. The authors should better motivate the choices of ‘lesions’ they apply to the ideal feedback. Without those motivations, the various feedback conditions considered seem arbitrary.

9) As noted by the authors, a more realistic version of the model is one where weight changes need to be inferred, as opposed to being fed to the network. How viable is the process of inferring weights from regressing the internal dynamics against cursor velocities in an actual network of neurons? It is not immediately clear what makes this strategy any more viable than the ideal feedback strategy. The fact that inferred weights deviate from the ideal weight changes is not sufficient to conclude that the former are more realistic.

Minor comments

1) Throughout the manuscript, one result that seems to come up at multiple occasions is the fact that a perfect manifold alignment is not required for performance in the task to max out. It is stated indeed that ~40% of overlap is generally enough between the readout and the intrinsic manifold to obtain high performance. To what do authors attribute this finding? Do they think it is also the case in a real BCI setting? How general is this number of 40%?

2) Currently the caption of Figure 4G does not explain what the black line corresponds to (it is only explained in the method section). Please add it in the caption for clarity.

3) Regarding the effect of noisy feedback (Figure 3B): is the difference between WMP and OMP robust across features of the model? The fact that WMP does better compared to OMP only in an intermediate range of noise makes it questionable whether it is not just an oddity of the model under the current parameter values.

4) I suggest clarifying the section on the relationship between OMP learning and the dimensionality of neural dynamics. This part would benefit from stating explicitly the goal of the analysis, which is to analyze how OMP learning differs for varying degrees of dimensionality. Moreover, the overall logic of the argument developed in that section is hard to follow. If I understand correctly the argument, in a high-D system, the variance is more evenly distributed across dimensions, which means that dimensions are all roughly equivalent, and therefore do not suffer too much from shuffling. Whereas in low-D, because the space is highly anisotropic, shuffling does impact performance more dramatically. Finally, I am unsure what the following sentence means: “the inferred feedback weights for an outside-manifold perturbation in our main simulation setup always pointed in directions which are within the original manifold”.

5) Would the results hold if the target is given to RNN as a tonic input (which more directly match the experimental setting) and not as a pulse?

6) An important point noted in the paper is “… our study gives a new perspective on the general problem of credit assignment, proposing that correct feedback learning critically depends on the alignment between output transformation and internal manifold in a specific neural pathway.” I missed how this was shown in the paper.

7) The solution in the weight space for either type of learning is not unique. For example, the solution found with pseudoinverse may be different from regression feedback and other learning strategies. Therefore, it is unclear how weigh-change measure in Figure 2 is relevant to a more biological learning. This is quite evident in Figure 2 where there are several instances of high performance but low manifold overlap indicating that the solutions under different retraining schemes and perturbations may not converge to the same solution, and are thus not necessarily comparable. The results shown simply indicate that there exists a solution with similar distance but doesn't guarantee that those solutions are as easily reachable through all learning mechanisms. Maybe, in the space of reachable weight states the distance is a lot longer for off-manifold. This needs some thinking and discussion.

8) The text is really opaque with respect to P-OMP. In fact, there is no mention of this despite it reappearing several times in figures.

Reviewer #2: Animal experiments show that in a BCI task outside-manifold remapping (OMR) are much difficult (if not impossible) to learn than within-manifold remapping (WMR) (Sadtler et al. 2014). In this manuscript authors address the issue of why there maybe a difference in the OMR and WMR. This question has been previously addressed by Waernberg and Kumar (PloS Comp 2019), where they showed that OMR may require a larger change in local connectivity than WMR. Similarly Menendez and Latham (2019) argued that the learning difficulty can be attributed to inputs. In this study authors provide a different explanation.

They have uses a recurrent rate network (RNN) that learns with an online recursive-least-square algorithm. In its initial formulation this algorithm requires the pre-existence of carefully tuned feedback weights, (not biologically plausible). The authors address this by showing that a) these feed-back can be noisy or sparse (Fig 3), b) the feed-back weights can be learned (Fig 4) and c) that a similar result can be found also with a local learning rule (Fig S6). They argue that only when feedback error signal is noisy that OMR is difficult. They show that when ideal-observer feedback is available there is no fundamental difference between WMR and OMR. When such ideal feedback is not available (as would be the case in a biologically plausible scenario), WMR outperforms OMR -- i.e. WMR is more robust to noise in the error signal.

Whether "it is possible to infer useful feedback weights by regressing neural activity against cursor dynamics" in the brain, is another discussion -- but I think this paper points to a variable that may control learning in neural networks -- at least those using supervised learning.

While the manuscript is very well written and illustrated with excellent graphics, there are few points that need to be addressed in a revised version.

- In contrast to the conclusion of Waernberg+Kumar paper, the present study shows that there is no difference in amount of local network rewiring required for OMR and WMR. After having read the paper I still do not understand why this should be the case? Unfortunately the current manuscript does not provide any intuitive or conceptual understanding of this. I wonder if authors can provide a more intuitive way to think about this. Authors try to address this question on page 8. But I don't understand why 'wrong Eigenvectors' should be qualitatively different for low and high dimensional dynamics? I think point needs some more explanation.

- In Fig S4 (which I think should be in the main text) author show the weights before and after BCI perturbations. Here authors show the effect rank of the dW matrix in different conditions. But I don't think it makes sense to compare the effect rank of dW with sparse rank-1 matrix. I think the comparison should be made with a sparse rank-2 because the task (and therefore dW) has 2 dimensions. My guess is that the effect rank of such a matrix would be comparable to that if dW_OMR and dW_WMR.

- It would be fair to mention the Waernberg and Kumar 2019 and Menendez and Latham (2019) already in the introduction. After all they have addressed the question by varying the local connectivity (Waernberg and Kumar) and by changing the inputs (Menedndez and Latham).

- Authors wrote that "one fundamental difference compared to Waernberg and Kumar is that, in our study, the weight dynamics are not constrained to one dimension (p. 13)." This statement is not quite correct, In Waernberg+Kumar paper the weight ”dynamics” was constrained to D dimensions where D is the dimensionality (> 1) of the intrinsic manifold and in sparse matrices, the matrix rank was much higher than the dynamics.

- Predictions: The first prediction says "for within-manifold perturbations, it is possible to infer useful feedback weights by regressing neural activity against cursor dynamics" This is not really a prediction -- this is what authors have shown to be the case in their model. Evidently, the model has many non-biological features including the learning rule, so just because something is seen in an RNN does not mean it is a prediction for the brain. Also, the second prediction should be reformulated in view of the paper by Oby et al. (2020).

- How was the MSE scaled? In figure S2A and S2E it looks like a typical error would be on the order of 0.2 for untrained networks. In Fig 2B and similar figures the error is around 1.0. In Fig S3 the scale is different for panels C and D, and different from the scale in the main figures?

- The sentence ”In order to assure that both types of perturbations cause approximately the same performance error we calculate the performance for 200 random perturbations of each type and then pick the ones which closely match in the performance error they cause.” is a bit unclear. How was this matching done? And does this mean the MSEs for WMP and OMP were equal by design in for example Fig 2B? Were these errors similar on average even without this matching?

- In the current formulation, the network equations are deterministic, i.e. has no noise. How important is this?

Minor:

- It is unclear whether the performance error e^P is a scalar or a vector. In the text, it says that is a mean squared error, suggesting that it is a scalar. But in the equation e = W^{fb}e^P it seems it should be a 2D vector, because W^{fb} is the pseudo-inverse of the 2xN read-out matrix T.

- I think a minus sign missing in the equation for dP? Laje & Buonomano (2013), Nicola & Clopath (2017) and Waernberg & Kumar (2019) all have a minus sign in the corresponding equation.

- There is no index j on the right-hand side of the equation for dW. I’m guessing one of the $I$s in the nominator should be a j?

- The notation dW and dP might suggest a continuous rate of change (like d/dt) rather than a step-wise update. Perhaps use ∆ instead of d if it’s a fixed update happening every (second) time step?

- The variable T is used for both trial duration and BCI transformation.

- The forward slash in the x-label of Fig 2C might be read as division at first glance.

**Have all data underlying the figures and results presented in the manuscript been provided?**

Reviewer #1: Yes

Reviewer #2: **No: **It will provided on request

PLOS authors have the option to publish the peer review history of their article (what does this mean?). If published, this will include your full peer review and any attached files.

Reviewer #1: No

Reviewer #2: No
---

## [Decision Letter · Decision Letter 1]

8 Dec 2020

Dear Dr. Clopath,

We are pleased to inform you that your manuscript 'Neural manifold under plasticity in a goal driven learning behaviour' has been provisionally accepted for publication in PLOS Computational Biology.

Best regards,

Abigail Morrison

Associate Editor

PLOS Computational Biology

Lyle Graham

Deputy Editor

PLOS Computational Biology

Reviewer's Responses to Questions

**Comments to the Authors:**

Reviewer #1: The authors have adequately addressed my comments. It was a pleasure reviewing this manuscript.

Reviewer #2: I am happy with the revision. All my concerns have been adequately addressed.

**Have all data underlying the figures and results presented in the manuscript been provided?**

Reviewer #1: Yes

Reviewer #2: Yes

PLOS authors have the option to publish the peer review history of their article (what does this mean?). If published, this will include your full peer review and any attached files.

Reviewer #1: **Yes: **Mehrdad Jazayeri

Reviewer #2: No

---

## [Editor Report · Acceptance letter]

15 Jan 2021

PCOMPBIOL-D-20-00993R1 

Neural manifold under plasticity in a goal driven learning behaviour

Dear Dr Clopath,

I am pleased to inform you that your manuscript has been formally accepted for publication in PLOS Computational Biology. Your manuscript is now with our production department and you will be notified of the publication date in due course.

With kind regards,

Jutka Oroszlan
